# Self-Consistent Models and Values

**Gregory Farquhar**
DeepMind

**Kate Baumli**
DeepMind

**Zita Marinho**
DeepMind

**Angelos Filos**
University of Oxford

**Matteo Hessel**
DeepMind

**Hado van Hasselt**
DeepMind

**David Silver**
DeepMind

## Abstract

Learned models of the environment provide reinforcement learning (RL) agents with flexible ways of making predictions about the environment. In particular, models enable planning, i.e. using more computation to improve value functions or policies, without requiring additional environment interactions. In this work, we investigate a way of augmenting model-based RL, by additionally encouraging a learned model and value function to be jointly *self-consistent*. Our approach differs from classic planning methods such as Dyna, which only update values to be consistent with the model. We propose multiple self-consistency updates, evaluate these in both tabular and function approximation settings, and find that, with appropriate choices, self-consistency helps both policy evaluation and control.

## 1 Introduction

Models of the environment provide reinforcement learning (RL) agents with flexible ways of making predictions about the environment. They have been used to great effect in planning for action selection [45, 29, 49], and for learning policies or value functions more efficiently [51]. Learning models can also assist representation learning, serving as an auxiliary task, even if not used for planning [25, 48]. Traditionally, models are trained to be consistent with experience gathered in the environment. For instance, an agent may learn maximum likelihood estimates of the reward function and state-transition probabilities, based on the observed rewards and state transitions. Alternatively, an agent may learn a model that only predicts behaviourally-relevant quantities like rewards, values, and policies [47].

In this work, we study a possible way to augment model-learning, by additionally encouraging a learned model $\hat{m}$ and value function $\hat{v}$ to be jointly *self-consistent*, in the sense of jointly satisfying the Bellman equation with respect to $\hat{m}$ and $\hat{v}$ for the agent's policy $\pi$. Typical methods for using models in learning, like Dyna [51], treat the model as a fixed best estimate of the environment, and only update the value to be consistent with the model. Self-consistency, by contrast, jointly updates the model and value to be consistent with each other. This may allow information to flow more flexibly between the learned reward function, transition model, and approximate value. Since the true model and value are self-consistent, this type of update may also serve as a useful regulariser.

We investigate self-consistency both in a tabular setting and at scale, in the context of deep RL. There are many ways to formulate a model-value update based on the principle of self-consistency, but we find that naive updates may be useless, or even detrimental. However, one variant based on a semi-gradient temporal difference objective can accelerate value learning and policy optimisation. We evaluate different search-control strategies (i.e. the choice of states and actions used in the self-consistency update), and show that self-consistency can improve sample efficiency in environments such as Atari, Sokoban and Go. We conclude with experiments designed to shed light on the mechanisms by which our proposed self-consistency update aids learning.

35th Conference on Neural Information Processing Systems (NeurIPS 2021).

## 2 Background

We adopt the Markov Decision Process (MDP) formalism for an *agent* interacting with an *environment* [7, 42]. The agent selects an action $A_t \in \mathcal{A}$ in state $S_t \in \mathcal{S}$ on each time step $t$, then observes a reward $R_{t+1}$ and transitions to the successor state $S_{t+1}$. The environment dynamics is given by a "true" model $m^* = (r, P)$, consisting of reward function $r(s, a)$ and transition dynamics $P(s'|s, a)$.

The behaviour of the agent is characterized by a policy $\pi : \mathcal{S} \to \Delta^{\mathcal{A}}$, a mapping from states to the space of probability distributions over actions. The agent's objective is to update its policy to maximise the *value* $v^\pi(s) = \mathbb{E}_{m^*, \pi} \left[ \sum_{j=0}^\infty \gamma^j r(S_{t+j}, A_{t+j}) \mid S_t = s \right]$ of each state. We are interested in model-based approaches to this problem, where an agent uses a learned *model* of the environment $\hat{m} = (\hat{r}, \hat{P})$, possibly together with a learned value function $\hat{v}(s) \approx v^\pi$, to optimise $\pi$.

The simplest approach to learning such a model is to make a maximum likelihood estimate of the true reward and transition dynamics, based on the agent's experience [32]. The model can then be used to perform updates for arbitrary states and action pairs, entirely in imagination. Dyna [51] is one algorithm with this property, that has proven effective in improving the data efficiency of RL [22, 31].

An alternative model-learning objective is *value equivalence*[1] i.e. equivalence of the true and learned models in terms of the induced values for the agent's policy. This can be formalised as requiring the induced Bellman operator of the model to match that of the environment:

$$\mathcal{T}_{m^*}^\pi v^\pi = \mathcal{T}_{\hat{m}}^\pi v^\pi, \tag{1}$$

where the Bellman operator for a model $\hat{m}$ is defined by $\mathcal{T}_{\hat{m}}^\pi v = \mathbb{E}_{\hat{m}, \pi} [r(s, a) + \gamma v(s')]$. Such models do not need to capture every detail of the environment, but must be consistent with it in terms of the induced Bellman operators applied to certain functions (in this case, the agent's value $v^\pi$).

We take a particular interest in value equivalent models for two reasons. First, they are used in state-of-the-art agents, as described in the next section and used as baselines in our deep RL experiments. Second, these algorithms update the model using a value-based loss, like the self-consistency updates we will study in this work. We describe this connection further in Section 3.3.

### 2.1 Value equivalent models in deep RL

A specific flavour of value equivalent model is used by the MuZero [47] and Muesli [25] agents. Both use a representation function (encoder) $h_\theta$ to construct a state $z_t = h_\theta(o_{1:t})$ from a history of observations $o_{1:t}$[2]. We define a deterministic recurrent model $\hat{m} \equiv m_\theta$ in this latent space. On each step $k$ the model, conditioned on an action, predicts a new latent state $z_t^k$ and reward $\hat{r}_t^k$; the agent also predicts from each $z_t^k$ a value $\hat{v}_t^k$ and a policy $\hat{\pi}_t^k$. We use superscripts to denote time steps in the model: i.e., $z_t^k$ is the latent state after taking $k$ steps with the model, starting in the root state $z_t^0 \equiv z_t$.

Muesli and MuZero unroll such a model for $K$ steps, conditioning on a sequence of actions taken in the environment. The learned components are then trained end-to-end, using data gathered from interaction with the environment, $\mathcal{D}_\pi^*$, by minimizing the loss function

$$\mathcal{L}_t^{\text{base}}(\hat{m}, \hat{v}, \hat{\pi} | \mathcal{D}_\pi^*) = \mathbb{E}_{\mathcal{D}_\pi^*} \sum_{k=0}^K \left[ \ell^r(r_{t+k}^{\text{target}}, \hat{r}_t^k) + \ell^v(v_{t+k}^{\text{target}}, \hat{v}_t^k) + \ell^\pi(\pi_{t+k}^{\text{target}}, \hat{\pi}_t^k) \right]. \tag{2}$$

This loss is the sum of a reward loss $\ell^r$, a value loss $\ell^v$ and a policy loss $\ell^\pi$. Note that there is no loss on the latent states $z_t^k$: e.g. these are not required to match $z_{t+k} = h_\theta(o_{1:t+k})$. The targets $r_{t+k}^{\text{target}}$ for the reward loss are the true rewards $R_{t+k}$ observed in the environment. The value targets are constructed from sequences of rewards and $n$-bootstrap value estimates $v_{t+k}^{\text{target}} = \sum_{j=1}^n \gamma^{j-1} r_{t+k+j} + \gamma^n \tilde{v}_{t+k+n}$. The bootstrap value estimates $\tilde{v}$, as well as the policy targets, are constructed using the model $\hat{m}$. This is achieved in MuZero by applying Monte-Carlo tree search, and, in Muesli, by using one-step look-ahead to create MPO-like [1] targets. Muesli also employs a policy-gradient objective which does not update the model. We provide some further details about Muesli in Appendix B.2, as we use

---

[1]We borrow the terminology from Grimm et al. [19], who provided a theoretical framework to reason about such models. Related ideas were also introduced under the name 'value aware' model learning [15, 14]

[2]We denote sequences of outcomes of random variables $O_1 = o_1, \ldots, O_t = o_t$ as $o_{1:t}$ for simplicity.

**Algorithm 1:** Model-based RL with joint grounded and self-consistency updates.

---

**input** : initial $\hat{m}, \hat{v}, \pi$
**output** : estimated value $\hat{v} \approx v^\pi$ and/or optimal policy $\pi^*$
**repeat**
> Collect $\mathcal{D}_\pi^*$ from $m^*$ following $\pi$
> Compute grounded loss: $\mathcal{L}^{\text{base}}(\hat{m}, \hat{v}, \pi | \mathcal{D}_\pi^*)$          `// e.g. Eq.`(2)
> Generate $\hat{\mathcal{D}}_\mu$ from $\hat{m}$ following $\mu$
> Compute self-consistency loss: $\mathcal{L}^{sc}(\hat{m}, \hat{v} | \hat{\mathcal{D}}_\mu)$      `// see Eqs.`(4,5,6)
> Update $\hat{m}, \hat{v}, \pi$ by minimising (e.g., with SGD): $\mathcal{L} = \mathcal{L}^{\text{base}} + \mathcal{L}^{sc}$

**until** *convergence*;

---

it as a baseline for our deep RL experiments. For comprehensive descriptions of MuZero and Muesli we refer to the respective publications [47, 25].

In both cases, the model is used to construct targets based on multiple *imagined* actions, but the latent states whose values are updated by optimising the objective (2) always correspond to *real* states that were actually encountered by the agent when executing the corresponding action sequence in its environment. Value equivalent models could also, in principle, update state and action pairs entirely in imagination (similarly to Dyna) but, to the best of our knowledge, this has not been investigated in the literature. Our proposed self-consistency updates provide one possible mechanism to do so.

## 3   Self-consistent models and values

A particular model $\hat{m}$ and policy $\pi$ induce a corresponding value function $v_{\hat{m}}^\pi$, which satisfies a Bellman equation $\mathcal{T}_{\hat{m}}^\pi v_{\hat{m}}^\pi = v_{\hat{m}}^\pi$. We describe model-value pairs which satisfy this condition as *self-consistent*. Note that the true model $m^*$ and value function $v^\pi$ are self-consistent by definition.

If an approximate model $\hat{m}$ and value $\hat{v}$ are learned independently, they may only be *asymptotically* self-consistent, in that the model is trained to converge to the true model, and the estimated value to the true value. Model-based RL algorithms, such as Dyna, introduce an explicit drive towards consistency throughout learning: the value is updated to be consistent with the model, in addition to the environment. However, the model is only updated to be consistent with transitions in the real environment $\mathcal{D}_\pi^*$, and not to be consistent with the approximate values. Instead, we propose to update both the model and values so that they are self-consistent with respect to trajectories $\hat{\mathcal{D}}_\mu$ sampled by rolling out a model $\hat{m}$, under action sequences sampled from some policy $\mu$ (with $\mu \neq \pi$ in general).

We conjecture that allowing information to flow more freely between the learned rewards, transitions, and values, may make learning more efficient. Self-consistency may also implement a useful form of regularisation – in the absence of sufficient data, we might prefer a model and value to be self-consistent. Finally, in the context of function approximation, it may help representation learning by constraining it to support self-consistent model-value pairs. In this way, self-consistency may also be valuable as an auxiliary task even if the model is not used for planning.

### 3.1   Self-consistency updates

For the model and value to be eventually useful, they must still be in some way grounded to the true environment: self-consistency is *not* all you need. We are therefore interested in algorithms that include both grounded model-value updates, and self-consistency updates. This may be implemented by alternating the two kind of updates, or in single optimisation step (c.f. Algorithm 1) by gathering both real $\mathcal{D}_\pi^*$ and imagined $\hat{\mathcal{D}}_\mu$ experience and then jointly minimize the loss

$$\mathcal{L}(\hat{m}, \hat{v}, \pi | \mathcal{D}_\pi^* + \hat{\mathcal{D}}_\mu) = \mathcal{L}^{\text{base}}(\hat{m}, \hat{v}, \pi | \mathcal{D}_\pi^*) + \mathcal{L}^{sc}(\hat{m}, \hat{v} | \hat{\mathcal{D}}_\mu). \tag{3}$$

There are many possible ways to leverage the principle of self-consistency to perform the additional update to the model and value ($\mathcal{L}^{sc}$). We will first describe several possible updates based on 1-step temporal difference errors computed from $K$-step model rollouts following $\mu = \pi$, but we will later

consider more general forms. The most obvious update enforces self-consistency directly:

$$\mathcal{L}^{\text{sc-residual}}(\hat{m}, \hat{v}) = \mathbb{E}_{\hat{m}, \pi} \left[ \sum_{k=0}^{K} \left( \hat{r}(s_k, a_k) + \gamma \hat{v}(s_{k+1}) - \hat{v}(s_k) \right)^2 \right]. \tag{4}$$

When used only to learn values, updates that minimize this loss are known as "residual" updates. These have been used successfully in model-free and model-based deep RL [57], even without addressing the double sampling issue [4]. However, it has been observed in the model-free setting that residual algorithms can have slower convergence rates than the TD(0) update [4], and may fail to find the true value [35]. In our setting, the additional degrees of freedom, due to the loss depending on a *learned* model, could allow the system to more easily fall into degenerate self-consistent solutions (e.g. zero reward and values everywhere). To alleviate this concern, we can also design our self-consistent updates to mirror a standard TD update, by treating the entire target $\hat{r}(s_k, a_k) + \gamma \hat{v}(s_{k+1})$ as fixed:

$$\mathcal{L}^{\text{sc-direct}}(\hat{m}, \hat{v}) = \mathbb{E}_{\hat{m}, \pi} \left[ \sum_{k=0}^{K} \left( \perp \left( \hat{r}(s_k, a_k) + \gamma \hat{v}(s_{k+1}) \right) - \hat{v}(s_k) \right)^2 \right], \tag{5}$$

where $\perp$ indicates a suppressed dependence (a "stop gradient" in the terminology of automatic differentiation). This is sometimes referred to as using the "direct" method, or as using a semi-gradient objective [52]. Dyna minimises this objective, but only updates the parameters of the value function (Fig. 1a). We propose to optimise the parameters of both the model and the value (Fig. 1b).

For $k = 0$, this will not update the transition model, as $v(s_0)$ does not depend on it. This necessitates the use of multi-step model rollouts ($k \geq 1$). Using multi-step model rollouts is also typically better in practice [27], at least for traditional model-based value learning. This form of the self-consistency update will also never update the reward model. We hypothesise this may be a desirable property, since the grounded learning of the reward model is typically well-behaved. Further, if the grounded model update enforces value equivalence with respect to the current policy's value, the transition dynamics are policy-dependent, but the reward model may be stationary. Consequently, we expect this form of update may have a particular synergy with the value-equivalent setting.

Alternatively, we consider applying a semi-gradient in the other direction:

$$\mathcal{L}^{\text{sc-reverse}}(\hat{m}, \hat{v}) = \mathbb{E}_{\hat{m}, \pi} \left[ \sum_{k=0}^{K} \left( \hat{r}(s_k, a_k) + \gamma \hat{v}(s_{k+1}) - \perp \left( \hat{v}(s_k) \right) \right)^2 \right] \tag{6}$$

This passes information forward in model-time from the value to the reward and transition model, even if $k = 0$.

### 3.2 Self-consistency in a tabular setting

First, we instantiate these updates in a tabular setting, using the pattern from Algorithm 1, but alternating the updates to the model and value for the grounded and self-consistency losses rather than adding the losses together for a single joint update. At each iteration, we first collect a batch of transitions from the environment $(s, a, r, s') \sim m^*$. The reward model is updated as $\hat{r}(s, a) \leftarrow r(s, a) + \alpha_r(r(s, a) - \hat{r}(s, a))$. We update the value with TD(0): $\hat{v}(s) \leftarrow (1 - \alpha_v)\hat{v}(s) + \alpha_v(r(s, a) + \gamma \hat{v}(s'))$. Here, $\alpha_r$ and $\alpha_v$ are learning rates.

Next, we update the model, using either a maximum likelihood approach or the value equivalence principle. The transition model $\hat{P}$ is parameterised as a softmax of learned logits: $\hat{P}(s'|s, a) = \text{softmax}(\hat{p}(s, a))[s']$. To learn a maximum likelihood model we follow the gradient of $\log \hat{P}(s'|s, a)$. For a value equivalent model, we descend the gradient of the squared difference of next-state values under the model and environment: $(\sum_{\hat{s}', a} \pi(a|s)\hat{P}(s'|s, a)\hat{v}(\hat{s}') - \hat{v}(s'))^2$.

Then, we update either the value only to be consistent with the model (Dyna), or update both the model and value to be self-consistent according to the objectives in the previous section. For control, we can construct an action-value estimate $\hat{Q}(s, a) = \hat{r}(s, a) + \gamma \sum_{s'} \hat{P}(s, a, s')\hat{v}(s')$ with the model $\hat{m} = \{\hat{r}, \hat{P}\}$, and act $\epsilon$-greedily using $\hat{Q}$.

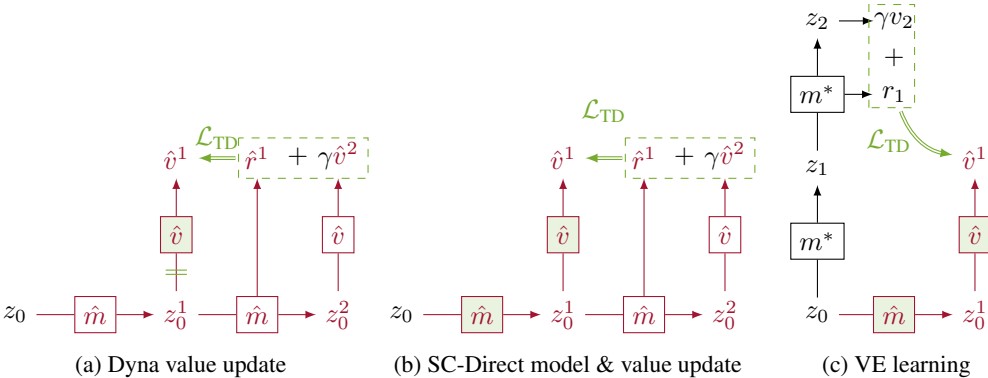

(a) Dyna value update      (b) SC-Direct model & value update      (c) VE learning

Figure 1: Schematic of model and/or value updates for $k = 1$ of a multi-step model rollout. Model predictions are red; dashed rectangle identifies the TD targets; superscripts denote steps in the model rollout. Real experience is in black and subscripted with time indices. Blocks represent functions; when color-filled they are updated by minimising a TD objective. (a,b) show planning updates that use only trajectories generated from the model: (a) Dyna updates only the value predictions to be consistent with the model; (b) our SC-Direct jointly updates both value *and model* to be self-consistent. (c) The value loss in the MuZero [47] form of VE learning is a grounded update that is similar in structure to SC, but uses real experience to compute the TD targets. The model unroll must therefore use the same actions that were actually taken in the environment $m^*$. A grounded update like (c) may be combined with updates in imagination like (a) or (b). Best viewed in color.

### 3.3 Self-consistency in deep RL

Self-consistency may be applied to deep RL as well, by modifying a base agent that learns a differentiable model $\hat{m}$ and value $\hat{v}$ using the generic augmented loss (3). We follow the design of Muesli [25] for our baseline, using the objective (2) to update the encoder, model, value, and policy in a grounded manner. We then add to the loss an additional self-consistency objective that generalises to the latent state representation setting:

$$\mathcal{L}_\pi^{sc} = \mathbb{E}_{z_0 \sim \mathcal{Z}_0} \mathbb{E}_{a^1,\dots,a^K \sim \mu} \left[ \sum_{k=1}^{K} l^v \big( \hat{G}_{k:K}^\pi, v(z^k) \big) \right]. \tag{7}$$

The base objective (2) already jointly learns a model and value to minimise a temporal-difference value objective $\ell^v(v^{\text{target}}, \hat{v})$. Our self-consistency update is thus closely related to the original value loss, but differs in two important ways. First, instead of using rewards and next states from the environment, the value targets $\hat{G}_{k:K}^\pi$ are $(K - k)$-step bootstrapped value estimates constructed with rewards and states predicted by rolling out the model $\hat{m}$. The reliance of MuZero-style VE on the environment for bootstrap targets is illustrated schematically in Figure 1c, in contrast to self-consistency. Second, because the targets do not rely on real interactions with the environment, we may use any starting state and sequence of actions, like in Dyna.

$\mathcal{Z}_0$ denotes the distribution of starting latent states from which to roll out the model for $K$ steps, and $\mu$ is the policy followed in the model rollout. When $\mu$ differs from $\pi$, we may use off-policy corrections to construct the target $\hat{G}_{k:K}^\pi$; in our experiments, we use V-Trace [13]. For now, we default to $\mu = \pi$, and to using the same distribution for $\mathcal{Z}_0$ as for the base objective. We revisit these choices in Section 4.2. Note that by treating different parts of the loss in 7 as fixed, we can recover the residual, direct, and reverse forms of self-consistency.

The self consistency objective is illustrated schematically in Figure 1. Subfigures (a) and (b) contrast a Dyna update to the value function with the SC-Direct update, which uses the same objective to update both value and model. Subfigure (c) shows the type of value equivalent update used in MuZero and Muesli. The objective is similar to SC-Direct, but the value targets come from real experience. Consequently, the model unroll must use the actions that were taken by the agent in the environment. Self-consistency instead allows to update the value and model based on any trajectories rolled out in the model; we verify the utility of this flexibility in Section 4.2.

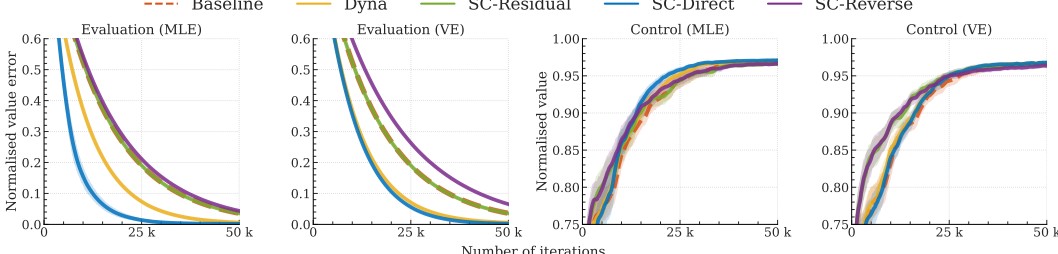

(a) Self-consistency in random tabular MDPs. Each experiment was run with 30 independent replicas using different random seeds. (Left) Normalised value prediction error for policy evaluation, using MLE and VE models respectively. (Right) Normalised policy values for control, using MLE and VE models respectively.

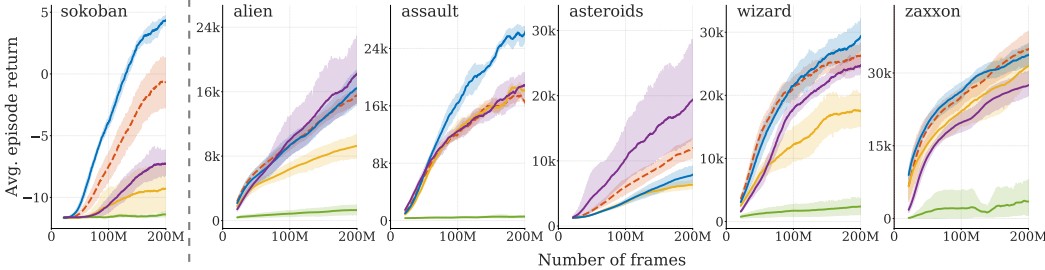

(b) Self-consistency on Sokoban and 5 Atari games. Each experiment was run with 5 independent replicas using different random seeds. The self-consistency updates were applied to variants of Muesli, a model-based RL agent using deep neural networks for function approximation and value equivalent models for planning.

Figure 2: An evaluation of Dyna and the self-consistency updates described in section 3.1. Each variant was evaluated in both a tabular setting (a) and with function approximation (b). The experiments included both MLE and value equivalent models. Shaded regions denote 90% CI.

## 4 Experiments

### 4.1 Sample efficiency through self-consistency

**Tabular.** In our first set of experiments, we used random "Garnet" MDPs [2] to study different combinations of grounded and self-consistent updates for approximate models and values. We followed an alternating minimization approach as described in Section 3.2. In each case, we applied the Dyna, SC-Direct, SC-Reverse, SC-Residual updates at each iteration starting at every state in the MDP. See Appendix A for further details of our experimental setup.

Figure 2a (on the left) shows the relative value error, calculated as $|(v(s) - \hat{v}(s))/v(s)|$, averaged across all states in the MDP, when evaluating a random policy. As we expected, Dyna improved the speed of value learning over a model-free baseline. We saw considerable difference among the self-consistent updates. The "residual" and "reverse" updates were ineffective and harmful, respectively. In contrast, the "direct" self-consistency update was able to accelerate learning, even compared to Dyna, for both MLE and value-equivalent models.

Figure 2a (on the right) shows the value of the policy (normalised by the optimal value), averaged across all states, for a control experiment with the same class of MDPs. At each iteration, we construct $\hat{Q}(s, a)$ estimates using the model, and follow an $\epsilon$-greedy policy with $\epsilon = 0.1$. In this case, the variation between methods is smaller. The "direct" method still achieves the best final performance. However, the "residual" and "reverse" self-consistency updates are competitive for tabular control.

**Function approximation.** We evaluated the same selection of imagination-based updates in a deep RL setting, using a variant of the Muesli [25] agent. The only difference is in the hyperparameters for batch size and replay buffer, as documented in the Appendix (this seemed to perform more stably as a baseline in some early experiments). All experiments were run using the Sebulba distributed

architecture [26] and the joint optimization sketched in Algorithm 1. We used 5 independent replicas, with different random seeds, to assess the performance of each update.

We evaluated the performance of the updates on a selection of Atari 2600 games [6], plus the planning environment Sokoban. In Sokoban, we omit the policy-gradient objective from the Muesli baseline because with the policy gradient, the performance ceiling for this problem is quickly reached. Using only the MPO component of the policy update relies more heavily on the model, giving us a useful testbed for our methods in this environment, but should not be regarded as state-of-the-art.

We first evaluated a Dyna baseline, using experience sampled from the latent-space value-equivalent model. As in the tabular case, we did so by taking the gradient of the objective (7) with respect to only the parameters of the value predictor $\hat{v}$. Unlike the simple tabular case, where Dyna performed well, we found this approach to consistently degrade performance at scale (Figure 2b, "Dyna"). To the best of our knowledge, this was the first attempt to perform Dyna-style planning with value-equivalent models, and it was a perhaps-surprising negative result. Note the contrast with the success reported in previous work [23] for using Dyna-style planning with MLE models on Atari.

Next we evaluated the self-consistency updates. We found that the "residual" self-consistency update performed poorly. We conjecture that this was due to the model converging on degenerate trivially self-consistent solutions, at the cost of optimising the grounded objectives required for effective learning. The degradation in performance was even more dramatic than in the tabular setting, perhaps due to the greater flexibility provided by the use of deep function approximation.

This hypothesis is consistent with our next finding: both semi-gradient objectives performed much better. In particular, the "direct" self-consistency objective increased sample efficiency over the baseline in all but one environment. We found the "reverse" objective to also help in multiple domains, although less consistently across the set of environments we tested. As a result, in subsequent experiments we focused on the "direct" update, and investigated whether the effectiveness of this kind of self-consistency update can be improved further.

## 4.2 Search control for self-consistency

*Search control* is the process of choosing the states and actions with which to query the model when planning [52]; we know from the literature that this can substantially influence the effectiveness of planning algorithms such as Dyna [40]. Since our self-consistency objective also allows flexibly updating the value of states that do not correspond to the states observed from real actions taken in the environment, we investigated the effect of different choices for imagination policy $\mu$ and starting states $\mathcal{Z}_0$.

**Which policy should be followed in imagination?** We now explore four choices for the policy followed in imagination to generate trajectories used by the self-consistency updates. The simplest choice is to sample actions according to the same policy we use to interact with the real environment ($\mu = \pi$). A second option is to pick a random action with probability $\epsilon$, and otherwise follows $\pi$ (we try $\epsilon$=0.5). The third option is to avoid the first action that was actually taken in the environment to ensure diversity from the real behaviour. After sampling a different first action, the policy $\pi$ is used to sample the rest (Avoid $a_0$). The last option is to replay the original sequence of actions that were taken in the environment (Original actions). This corresponds to falling back to the base agent's value loss, but with rewards and value targets computed using the model in place of the real experience.

In Figure 3a, we compare these options. A notable finding was that just replaying exactly the actions taken in the environment (in brown) performed considerably worst than re-sampling the actions from $\pi$ (in blue). This confirms our intuition that the ability to make use of arbitrary trajectories is a critical aspect of the self-consistency update. Introducing more diversity by adding noise to the policy $\pi$ had little effect on most environments, although the use of $\epsilon = 0.5$ did provide a benefit in `assault` and `asteroids`. Avoiding the first behaviour action, to ensure that only novel trajectories are generated in imagination, also performed reasonably well. Overall, as long as we generated a new trajectory by sampling from a policy, the self-consistency update was fairly robust to all the alternatives considered.

**From which states should we start imagined rollouts?** A search control strategy must also prescribe how the starting states are selected. Since we use a latent-space model, we investigated using latent-space perturbations to select starting states close (but different) from those encountered in the environment. We used a simple perturbation, multiplying the latent representation $z_t^0$ element-wise

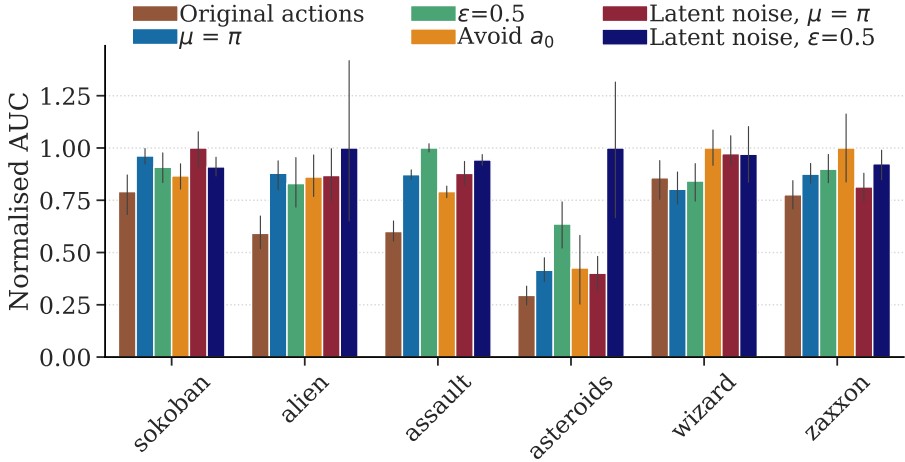

(a) Different search control strategies for self-consistency. Each experiment used 5 random seeds, error bars denote 90% CI.

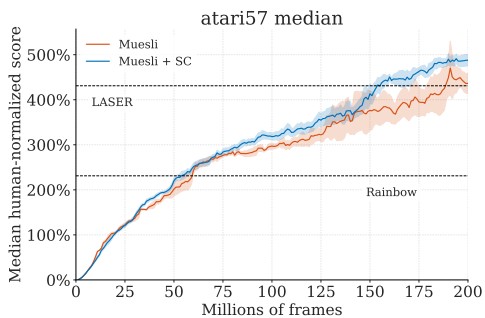

(b) Median human-normalised episode return on across 57 Atari games. Each experiment used 4 random seeds, shaded region denote 90% CI.

(c) Performance on 9x9 self-play Go, with evaluation against Pachi [5] Each experiment used 5 random seeds, shaded region denote 90% CI.

Figure 3: Search control for self consistency, and evaluations on Atari and Go.

by noise drawn from $\mathcal{U}(0.8, 1.2)$ (Latent noise, $\mu = \pi$). The results for these experiments are also shown in Figure 3a. The effect was not large, but the variant combining latent state perturbations with $\epsilon-$ exploration in the imagination policy (Latent noise, $\epsilon = 0.5$) performed best amongst all variants we considered. The fact that some benefit was observed with such rudimentary data augmentation suggests that diverse starting states may be of value. Self-consistency could therefore be effective in semi-supervised settings where we have access to many valid states (e.g. reachable configurations of a robotic system), but few ground-truth rewards or transitions.

**A full evaluation.** In the next experiments we evaluated our self-consistency update on the full Atari-57 benchmark, with the search-control strategy found to be most effective in the previous section. As shown in Figure 3b, we observed a modest improvement over the Muesli baseline in the median human-normalised score across the 57 environments. We also evaluated the same self-consistency update in a small experiment on self-play 9x9 Go in combination with a Muesli-MPO baseline, with results shown in Figure 3c. Again, the agent using self-consistency updates outperformed the baseline, both with (solid lines) and without (dashed lines) the use of MCTS at evaluation time. This confirms that the self-consistent model is sufficiently robust to be used effectively for planning online.

### 4.3 How does self-consistency work?

**Representation learning.** In our final set of experiments we investigated how self-consistency affects the learning dynamics of our agents. First, we analysed the impact of self-consistency on the learned representation. In Figure 4a we compare (1) a policy-gradient baseline "PG" that does not use a model (2) learning a value-equivalent model purely as an auxiliary task "+VE Aux", and (3)

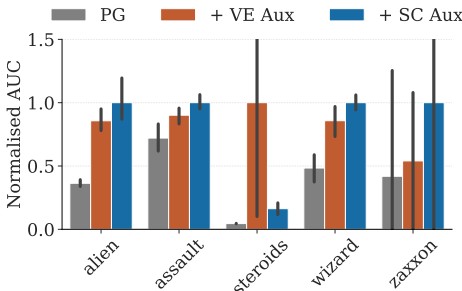
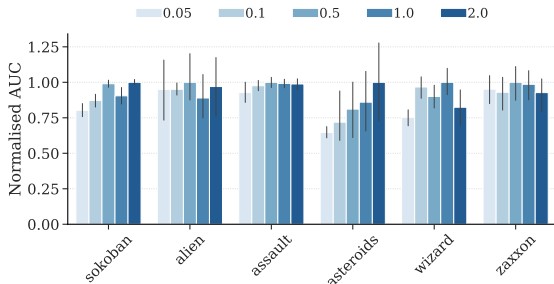

(a) Policy gradient baseline (PG), adding VE as an auxiliary task (+VE Aux) improves the representation, adding SC improves even further (+SC Aux).

(b) Episode return area under curve as a function of the ratio between number of imagined trajectories and real trajectories in each update to the model and value parameters.

Figure 4: Self-consistency can help representation learning as an auxiliary task.

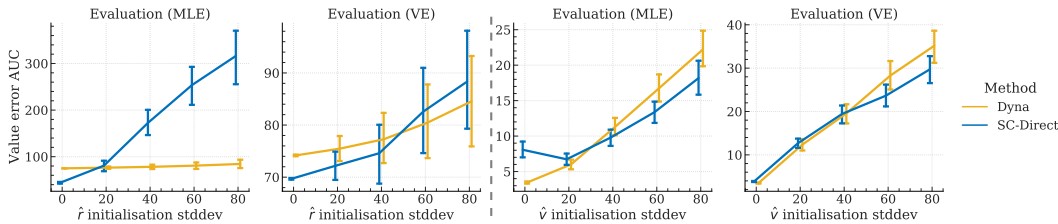

Figure 5: Tabular policy evaluation error as the initialisation of reward (left) and value (right) are varied. Gaussian noise is added to the true $r$ or $v^\pi$ to initialise $\hat{r}$ and $\hat{v}$. Error bars show $90\%$CI.

augmenting this auxiliary task with our self-consistency objective "+SC Aux". When we configure Muesli to use the model purely as an auxiliary there is no MPO-like loss term, and the value function bootstrap uses state values rather than model-based $Q$ values. We found that self-consistency updates were helpful, even though the model was neither used for value estimation nor policy improvement; this suggests that self-consistency may aid representation learning.

**Information flow.** Self-consistency updates, unlike Dyna, move information between the reward model, transition model, and value function. To study this effect, we returned to tabular policy evaluation. We analysed the robustness of self consistency to initial additive Gaussian noise in the reward model. Figure 5 (left) shows the area under the value error curve as a function of the noise. With worse initialisation the effectiveness of the self-consistency update deteriorates more rapidly than the Dyna update, especially when learning with MLE. The reward model is learned in a grounded manner, so will receive the same updates with Dyna or SC. In the Dyna case, the poor reward initialisation can only pollute the value function. With SC, information can also flow from the impaired reward model into the dynamics model, and from there damage the value function further in turn. A different effect can be seen when varying the value initialisation by adding noise to the true value, as shown in Figure 5 (right). Since the value is updated by SC, it is possible for a poorly initialised value to be fixed more rapidly if SC is effective. Indeed, we see a slight trend towards a greater advantage for SC when value initialisation is worse. However, this is not a guarantee; certain poor initialisations could lead to self-reinforcing errors with SC, which is reflected in the overlapping confidence intervals.

**Scaling with computation.** Self-consistency updates, like Dyna, allow to use additional computation without requiring further interactions in the environment. In Figure 4b, we show the effect of rolling out different numbers of imagined trajectories to estimate the self-consistency loss, by randomly choosing a fraction of the states in each batch to serve as starting points (for ratios greater than one we randomly duplicated starting states). In the domains where self-consistency was beneficial, the potential gains often increased with the sampling of additional starting-states. However, in most cases saturation was reached when one imagined trajectory was sampled for each state in the batch (ratio=1 was the default in the experiments described above).

# 5 Related work

Model learning can be useful to RL agents in various ways, such as: (i) aiding representation learning [46, 28, 33, 20, 25] (ii) planning for policy optimisation and/or value learning [55, 51, 21, 10]; (iii) action selection via local planning [45, 49]. See Moerland et al. [36] or Hamrick et al. [24] for a survey of how models are used in RL. Many approaches to *learning* the model have been investigated in the literature, usually based on the principle of maximum likelihood estimation (MLE) [32, 51, 38, 21, 31, 10, 22]. Hafner et al. [23] notably showed that a deep generative model of Atari games can be used for Dyna-style learning of a policy and value only in the imagination of the model, performing competitively with model-free methods.

RL-informed objectives have been used in some recent approaches to learn implicit or explicit models that focus on the aspects of the environment that are relevant for control [53, 50, 39, 16, 30, 47, 18, 44, 25]. This type of model learning is connected to theoretical investigations of "value-aware" model learning [15], which discusses an equivalence of Bellman operators applied to worst-case functions from a hypothesis space; "iterative value-aware model learning" [14] which, closer to our setting, uses an equivalence of Bellman operators applied to a series of agent value-functions; and the "value equivalence" (VE) principle [19], which describes equivalences between Bellman operators applied to sets of arbitrary functions.

All the approaches to model learning discussed above leverage observed *real experience* for learning a model of the environment. In contrast, our self-consistency principle provides a mechanism for model learning using *imagined experience*. This is related to the work of Silver et al. [50], where self-consistency updates were used to aid learning of temporally abstract models of a Markov reward process. Filos et al. [17] also used self-consistency losses, for offline multi-task inverse RL. Self-consistency has been applied in other areas of machine learning as well. Consider, for instance, back-translation [8, 12] in natural language processing, or CycleGANs [58] in generative modelling.

In this paper we considered models that can be rolled forward. It is also possible to consider *backward* models, that can be used to assign credit back in time [54, 11]. The principle of self-consistency we introduce in this work can be extended to these kinds of models fairly straightforwardly. A different type of self-consistency was studied concurrently by Yu et al. [56], who develop a model-based system where a forward and backward model are trained to be cyclically consistent with each other.

# 6 Conclusion

We introduced the idea of *self-consistent* models and values. Our approach departs from classic planning paradigms where a model is learned to be consistent with the environment, and values are learned to be consistent with the model. Amongst possible variants, we identified as particularly promising an update modelled on semi-gradient temporal difference objectives. We found this update to be effective both in tabular settings and with function approximation. Self-consistency updates proved particularly well suited to deep value-equivalent model-based agents, where traditional algorithms such as Dyna were found to perform poorly. The self-consistency objectives discusses in this paper are based on a form of policy-evaluation; future work may investigate extensions to enable value-equivalent model-based agents to perform policy *improvement* for states in arbitrary trajectories drawn from the model.

## Acknowledgments and Disclosure of Funding

We would like to thank Ivo Danihelka, Junhyuk Oh, Iurii Kemaev, and Thomas Hubert for valuable discussions and comments on the manuscript. Thanks also to the developers of Jax [9] and the DeepMind Jax ecosystem [3] which were invaluable to this project. The authors received no specific funding for this work.

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
