# A Tabular Experiments

**Garnet MDP.** Our version of the Garnet MDP is implemented as follows. An unnormalised transition matrix is sampled from $\mathcal{U}^{|\mathcal{S}| \times |\mathcal{A}| \times |\mathcal{S}|}(0, 1)$. For each state, we choose uniformly 1 to $|\mathcal{S}|$ states to be used as successors, the rest have their corresponding entries in the transition matrix set to zero. The matrix is then normalised. The deterministic reward function is sampled from $\mathcal{N}^{|\mathcal{S}| \times |\mathcal{A}|}(0, 1)$.

**Hyperparameters.** The hyperparameters for our tabular experiments are given in Table 1. $\alpha_{\text{TD}}$ was chosen from $\{0.01, 0.03, 0.1, 0.3\}$ for achieving the lowest final value error for the model-free baseline. $\alpha_{\text{MLE}}$ and $\alpha_{\text{VE}}$ were chosen from $\{0.1, 0.3, 1.0, 3.0\}$ for achieving the lowest final value error for Dyna. $\alpha_{\text{SC-VE}}$ and $\alpha_{\text{SC-MLE}}$ were chosen from a multiplier $\{0.1, 0.3, 1.0, 3.0, 10.0\}$ of the base model learning rate, for achieving the lowest final value error over all SC variants.

Table 1: Tabular experiment parameters.

| PARAMETER | VALUE |
|---|---|
| $\|\mathcal{S}\|$ | 20 |
| $\|\mathcal{A}\|$ | 4 |
| Discount | 0.99 |
| $\epsilon$ (for control) | 0.1 |
| Batch size | 8 |
| $\hat{p}$ initialisation | logDirichlet($\mathbf{1}$) |
| $\hat{r}$ initialisation | $\mathcal{N}(0, 1)$ |
| $\hat{v}$ initialisation | $\mathcal{N}(0, 1)$ |
| $K$ | 2 |
| $\alpha_{\text{TD}}$ | 0.03 |
| $\alpha_{\text{MLE}}$ | 1.0 |
| $\alpha_{\text{VE}}$ | 3.0 |
| $\alpha_{\text{SC-MLE}}$ | 10.0 |
| $\alpha_{\text{SC-VE}}$ | 0.3 |

# B Deep RL Experiments.

## B.1 Environments

**Atari.** The Atari suite consists of various discrete-action video games. The Atari configuration used is described in Table 2, following the recommendations of Machado et al. [34].

Table 2: Atari parameters. In general, we follow the recommendations by Machado et al. [34].

| PARAMETER | VALUE |
|---|---|
| Random modes and difficulties | No |
| Sticky action probability $\varsigma$ | 0.25 |
| Start no-ops | 0 |
| Life information | Not allowed |
| Action set | 18 actions |
| Max episode length | 30 minutes (108,000 frames) |
| Observation size | $96 \times 96$ |
| Action repetitions | 4 |
| Max-pool over last N action repeat frames | 4 |
| Total environment frames, including skipped frames | 200M |

**Sokoban.** Sokoban is a single player puzzle game where a player's avatar can push N boxes (up down left or right) around a procedurally generated warehouse, attempting to get to a state where each box is situated on one of N target storage locations. Our Sokoban environment uses a 10x10 grid. Levels are generated using the procedure described by Racanière et al. [43].

**Go.** Go is a two player abstract-strategy board game where the goal is to surround more territory on a NxN (in our case 9x9) board by placing black or white stones on the intersections of the board. For our Go experiments we evaluated against Pachi [5] configured with 10,000 simulations.

## B.2 Muesli.

Our deep RL experiments are based on the Muesli algorithm. For convenience, we provide some of the algorithmic details here, but refer the reader to Hessel et al. [25] for a comprehensive description.

The reward, value, and policy losses from equation 2 use a cross-entropy loss. To do so, the reward and value function are represented by a categorical distribution over 601 linearly spaced bins covering the range -300 to 300. Scalar targets are transformed into a probability distribution by placing the appropriate weight on the two nearest bins (e.g. 1.3 is represented with $p(1) = 0.7, p(2) = 0.3$). Rewards and values also use the non-linear value transform introduced by Pohlen et al. [41]. The value targets are computed by using the model to estimate $Q$-values with one-step lookahead and using Retrace [37]. The policy targets are also computed with one-step $Q$-values. A state-value baseline is subtracted to compute advantages, which are clipped in [-1, 1], exponentiated, and normalised to produce a target $\pi^{\text{target}} = \pi_{\text{CMPO}}$ given by

$$\pi_{\text{CMPO}}(a|s) = \frac{\pi_{\text{prior}}(a|s) \exp\left(\text{clip}(a\hat{\text{d}}v(s, a), -c, c)\right)}{z_{\text{CMPO}}(s)}, \tag{8}$$

where $z_{\text{CMPO}}(s)$ is a normaliser such that $\pi_{\text{CMPO}}$ is a probability distribution.

Muesli also uses a policy gradient loss which does not update the model, using advantages estimated from rollouts. When stated in the main text, we disable this part of the loss to highlight the effect of self-consistency on the model.

## B.3 Experimental details.

We use a Sebulba distributed architecture [26] to run the deep RL experiments. Our hyperparameters are given in Table 2, following by default the choices of Muesli [25] except for the replay buffer size and proportion of replay in a batch. We found our hyperparameters more stable in some early investigations but did not comprehensively verify the impact of these changes. We use the smaller "Impala" network architecture also given by Hessel et al. [25]. For self-consistency, we did not use the moving-average target parameters (this was found to not have a large effect).

For Sokoban, we used a replay proportion of only 6/60 to accelerate the wall-clock time of experiments. For Go, we used a discount of -1, and online data only, in order to perform symmetric self-play. The additional changes to the hyperparameters for Go are shown in Table 4.

In our experiments on the representation learning effect of self-consistency (Section 5), we trained an actor-critic agent as a baseline. To do so we set the weight on $\ell^\pi$ and $\ell^r$ in equation 2 to zero, and computed $\ell^v$ only for $k = 0$, which does not use the model. For $v^{\texttt{target}}$ we bootstrap using value estimates without the one-step model expansion. This baseline does not use a model in any way. To use a VE model as an auxiliary task, we add back in $\ell^r$ and $\ell^v$ for $k > 0$. Note that Muesli uses separate parameters for $v$ at $k = 0$ and $k > 0$, so this value update does not directly affect the model-free $k = 0$ value which is used for the policy gradient estimate. Consequently, this is purely an auxiliary task. Similarly, the auxiliary self-consistency objective, when enabled, does not affect policy updates except through the shared representation.

The experiments reported in this paper required approximately 50k TPU-v3 device-hours. A comparable amount of computation was used for other preliminary investigations over the course of the project.

## B.4 Additional experimental data.

To compute a normalised AUC in the main text, we used the minimum episode return recorded for each environment to compute a minimum AUC, and the maximum average AUC over all methods as a maximum. In Figure 6, we include full learning curves for the experiments which were summarised by

Table 3: Default hyperparameters for deep RL experiments.

| HYPERPARAMETER | VALUE |
|---|---|
| Batch size | 60 sequences |
| Sequence length | 30 frames |
| Model unroll length | 5 |
| Replay proportion in a batch | 52/60 |
| Replay buffer capacity | 9,000,000 frames |
| Optimiser | Adam |
| Initial learning rate | $3 \times 10^{-4}$ |
| Final learning rate (linear decay) | 0 |
| Discount | 0.995 |
| Target network update rate $\alpha_{\text{target}}$ | 0.1 |
| Value loss weight | 0.25 |
| Reward loss weight | 1.0 |
| Policy gradient loss weight (Atari only) | 3.0 |
| Retrace $\mathbb{E}_{A\sim\pi}[\hat{q}_\pi(s,a)]$ estimator | 16 samples |
| KL$(\pi^{\text{CMPO}}, \pi)$ estimator | 16 samples |
| Variance moving average decay $\beta_{\text{var}}$ | 0.99 |
| Variance offset $\epsilon_{\text{var}}$ | $10^{-12}$ |
| SC unroll length $K$ | 3 |
| SC loss weight | 0.25 |

Table 4: Modified hyperparameters for 9x9 Go self-play experiments.

| HYPERPARAMETER | VALUE |
|---|---|
| Network architecture | MuZero net with 6 ResNet blocks |
| Batch size | 192 sequences |
| Sequence length | 49 frames |
| Replay proportion in a batch | 0% |
| Initial learning rate | $2 \times 10^{-4}$ |
| Target network update rate $\alpha_{\text{target}}$ | 0.01 |
| Discount | -1 (self-play) |
| Multi-step return estimator | V-trace |
| V-trace $\lambda$ | 0.99 |

AUC in the main text. Figure 6d also shows a "Model Only" SC update, to contrast with Dyna which only updates the value parameters. We found this performed comparably to a joint update of model and value, indicating the importance of updating model and representation with self-consistency in this setting.

We also investigated the effect of the length of the model rollout used in calculating the self-consistency loss. As in traditional model-based planning, we expect that if the model is rolled out too far, updates using it will become worse as errors compound. The results are shown in Figure 7, confirming that self-consistency degrades performance if applied over a longer unroll than used in the grounded model-learning update (5 steps in our experiments).

Original actions  ·····  $\mu = \pi$  ——  $\varepsilon=0.5$  ——  Avoid $a_0$  ——  Latent noise, $\mu=\pi$  ——  Latent noise, $\varepsilon=0.5$

(a) Avg Episode Return vs number of frames for sokoban (leftmost) and a subset of atari games.

····· 0.05  ——  0.1  ——  0.5  ——  1.0  ——  2.0

(b) Different rollout ratios of imagined vs. real trajectories.

····· PG  ——  + VE Aux  ——  + SC Aux

(c) SC as an auxiliary task.

········ Baseline  ——  Dyna  ——  SC-Direct (Model Only)  ——  SC-Direct  ——  SC +latent noise +$\varepsilon=0.5$

(d) Self-consistency results on Atari57 and Sokoban for different updates. Baseline does not impose SC, Dyna learns a SC value only, SC-Model learns a SC model only and SC-Direct and SC-Direct (Tuned) update both model and value.

Figure 6: Full learning curves for ablations summarised in the text.

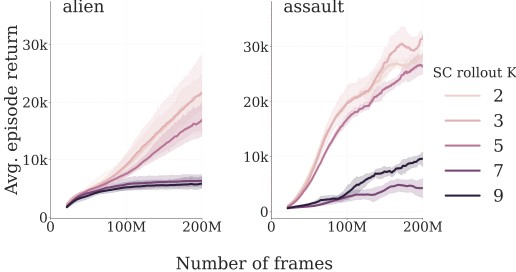

Figure 7: Varying the length K of the rollout used to compute the self-consistency loss with SC-Direct. Performance degrades after about 5 model unrolls (the grounded model-learning unrolls for 5 steps).