# OpenReview forum: "Self-Consistent Models and Values"
_NeurIPS.cc/2021/Conference — NeurIPS 2021 Poster_

### Official Review · Reviewer_TCCY · 2021-07-05

**Rating:** 7
**Confidence:** 4

**Summary:**

The paper offers a new approach to model-based reinforcement learning with their idea of self-consistency of models and values. In the traditional (Dyna) model-based RL paradigm, an approximate model is learned with ground truth transitions and values (and policies) are updated to be consistent with the learned model. The proposed self-consistency update however, formulates a new update to the approximate model so as to make it consistent with the current value estimates using completely imagined (virtual) experience. This update is inspired by minimizing the bellman residual of the approximate model and value for a given policy (since for the true value and model, the bellman equation is satisfied and the residual is zero). An empirical analysis is presented for a set of variations of the proposed self-consistency objective with two types of (grounded) model updates -- MLE (maximum-likelihood) and value equivalent updates. In both tabular and function approximate settings, improved sample efficiency is demonstrated by the proposed method.

**Limitations And Societal Impact:**

The authors have sufficiently addressed limitations of their work. For example, Section 4.3 demonstrates the effect of noise in reward or value estimates on their method in comparison to the Dyna baseline.

**Main Review:**

Overall the paper presents a novel self-consistency objective with a thorough empirical analysis of its variants on raw performance / sample efficiency (in tabular and deep RL settings), choices of search control (i.e. policy and states to use in imagined experience) and its benefits to representation learning. While not all experiments show a clear cut benefit of using self-consistency in all the cases visited in this paper, the unbiased analysis itself is an important contribution.
As a result, I feel that this paper in its current form is a significant and relevant contribution for the model-based RL community.

Below are more specific strengths and weaknesses.

## Strengths
- The paper proposes a self-consistency update which offers a new take on model updates in model-based RL. This update is important and distinct from most prior work in that it uses imagined (virtual) experience as opposed to ground truth (real) experience, allowing for freedom to update the model without collecting additional data from the true environment. This also leads to questions about what states and policy to choose for such updates and these questions were answered in the search control strategies section (Section 4.2).
- The self-consistency update is also motivated as a regularizer and shown to promote information “flow” in the presence of noisy inputs (Section 4.3 - Information Flow).
- Three different variations of the self-consistency updates are studied -- sc-residual, sc-direct and sc-reverse with a clear explanation of how each affects updates for the policy, reward or transition model. While sc-direct is the primary method, the ablations are also evaluated in the experiments providing valuable insights for future work in this area.
- The benefits of self-consistency to representation learning are insightful, especially given that the self-consistency is used as a purely auxiliary objective where the model was not used for value estimation or policy improvement.
- The evaluation of the method on Atari and Go show a clear sample efficiency improvement of the objective on top of the baseline methods.

## Weaknesses
- The motivation for using value equivalent model learning in addition to MLE for the model-update choice could be made more clear in the paper. Currently, I was able to understand the connection to value equivalence based on the fact that both self-consistency and value equivalence take inspiration from the bellman operator or bellman equation for their updates. However, the following statements were difficult to understand.
    - (a) Lines 76-78 (“Value equivalent models could also … in literature”) -- I have no idea how this can be achieved without further details. This sentence seems to support the inclusion of value equivalence as a major part of this paper so further details would help.
    - (b) Lines 124-126 (“Further, if grounded … value-equivalent setting”). It is not made clear how the policy dependence of the transition model can allow for synergy with the proposed self-consistency update. Is it because of the information flow? Further details on this would help as this is another important justification for the use of value equivalence.
    - Overall, the link between value equivalence and self-consistency is not explicitly addressed but rather is inferred after reading the above two sentences. It is also missing from the introduction. I would recommend addressing the link and the motivation for using value equivalence early on in the paper before existing value equivalent methods are introduced.

- The experiment on information flow in Section 4.3 is an important contribution despite the lack of statistically significant differences between Dyna and SC-Direct. However, I do not understand the justification for the poor reward estimates affecting self-consistency more while the poor value estimates affect Dyna more. The inversion of trend from reward noise to value noise is the issue here. The trends mostly have overlapping CIs other than the first figure, so I would recommend either dropping any strong claims on difference in performance or making the justifications for these trends more clear.

- Line 312 in conclusion (last sentence): This future work sentence is not clear. Why is there a need to perform policy improvement for value equivalent model-based agents in imagined experience? Don’t existing value equivalent model-based agents already perform policy improvement in this way e.g. [11]?


**Time Spent Reviewing:**

3

---

> ### Author Response · Authors · 2021-08-10
> **Thank you for your review**
>
> Thank you for your positive comments! We will take your feedback on board to improve the clarity of the writing.
>
> The connection of SC and VE is indeed somewhat subtle.
> The idea of using Dyna, or of using self-consistency as we propose, is broadly applicable in model-based RL, whether the model is learned with MLE, VE, or in some other way. This is the perspective we were trying to emphasise in the introduction, and in lines 76-78, where we note that VE models can be used in these ways (although our paper is to the best of our knowledge the first attempt to do so).
>
> However, we do focus more on VE-style models. The first reason is simply that, in the form of MuZero and Muesli, they represent state-of-the-art model-based agents.
> The second is as you describe: the VE and SC objectives are structurally similar, using the Bellman equation for their updates. This is the point we were hoping to clarify in Fig 1 by comparing (b) and (c).
> We will bring these points earlier into the text as you suggest in order to more clearly motivate the focus on VE models in our later experiments.
> The point in lines 124-126 is indeed related to the SC/VE similarity described above. VE and SC focus the model on being accurate and consistent with respect to the current policy, which is all that is needed for local policy improvements. In this sense there is a philosophical alignment between the methods, although this does not guarantee an empirical synergy. For both VE and SC-Direct, it further seems effective to treat reward-modeling as an independent supervised problem, which we alluded to in this part but should point out separately.
>
> We agree the language in the section on information flow can be softened to better reflect the overlapping CIs, and the hypothesis for the mechanism elaborated. The key difference between reward and value is that the reward function is learned only in a grounded manner; it is not updated by self-consistency. Consequently, poor reward initialisation leads only to a pollution of value in the Dyna case, and of value and transition model (likely worse!) in the SC case. In contrast, the value *is* updated by SC, so a poorly initialised value could be fixed more rapidly if SC is effective. This is by no means certain, however - one can imagine certain poor initialisations leading to self-reinforcing errors with SC. We believe this variance is reflected in the overlapping CIs, as you suggest.
>
> To elaborate on the final sentence in the conclusion: MuZero, Muesli, and IterVAML use imagined experience for policy improvement, but only to improve the policy for states that were visited, rather than for arbitrary states. Similarly to SC, we could do policy improvement using arbitrary trajectories drawn from the model, adding more flexibility and another mechanism for scaling with additional computation.

---

> > ### Comment · Reviewer_TCCY · 2021-08-18
> > **Thank you for the response and clarifications**
> >
> > I am happy with the proposed changes regarding the connection between SC and VE and also the claims/language around the overlapping CI results. Further, I appreciate the clarification regarding the future work sentence, it makes sense to me now.
> >
> > A tidbit regarding Figure 1 (need not be addressed now, I would recommend addressing this before the camera ready if accepted) -- I understand that it tries to distinguish between VE and SC and I recognize the effort put into it. However, it still comes off as a challenging figure to understand. First, it is not very "visual" -- in order to understand the symbols in the figure, the caption has to be read carefully and then the figure has to be carefully inspected again. Second, the color schemes are also not present in the figure itself, but rather in the captions. This makes the figure not self-contained -- the reader cannot glean relevant information from a glance of the figure alone. In my first read through, despite being knowledgable about value-equivalence, the figure still took quite a long time to process and understand. I would recommend using phrases instead of symbols and having a tiny legend in the figure that explains the color scheme -- essentially the figure should convey all relevant information without the caption and should have easy to grasp nomenclature for each node.

---

### Official Review · Reviewer_2igq · 2021-07-16

**Rating:** 6
**Confidence:** 3

**Summary:**

This paper considers the problem of model-based reinforcement learning. The authors introduce a new loss, which drives the learned model and value function to be consistent with each other, rather than the model being consistent with the real environment and the value function being consistent with the real environment and the learned model. Experiments show that the proposed self-consistency loss can improve the sample efficiency and final performance.

**Limitations And Societal Impact:**

I didn't find where the authors discuss the limitations.

**Main Review:**

The writing can be improved. The writing for method and intuition are reasonable, but for experiments, a lot of missing details make understanding it difficult.

It might be a good idea to introduce Muesli to readers who are not familiar with it. It's hard to understand what "using only the MPO component of the policy update" means without any proper introduction to Muesli. The paper mainly uses Muesli as its baseline while the experiments also use "a variant of the Muesli agent". I didn't find a detailed description of the variant. The lack of code, even implementable pseudocode, and implementation details, together with a heavy computational resources requirement, make reproducing this paper very difficult.

This paper only applies self-consistency to Muesli (and a grid world) in many domains with discrete control (Sobokan, Atari games, 9x9 self-play Go), while it seems to me that the self-consistent model/values are also applicable to other model-based RL algorithms in continuous control domain, e.g., MBPO.

Eq.7: How is G computed? Please give an explicit form rather than simply citing a paper if it's not elementary.

Sec 4.1: What's the model-free baseline? How's the value normalized?

> (L.174) In contrast, the “direct” self-consistency update was able to accelerate learning, even compared to Dyna, for both MLE and value-equivalent models.
I didn't see a significant advantage of SC-direct over Dyna with VE models.

Sec 4.1: Please provide a short description of the environments used (especially Sokoban).

Fig.6 (a): 6 curves in the legend, 4 curves in plots?
Fig.6 (d): The figure caption should match the titles of each subplot.

The improvement of Muesli + SC over Muesli seems to be minor.

Sec 4.2: Is the policy pi deterministic or stochastic?

> (L.257-259) We found that self-consistency updates were helpful, even though the model was neither used for value estimation nor policy improvement...
I don't understand. In both VE aux and SC aux, the model is used to improve the value function. What does "the model was neither used for value estimation" mean?

L.273: "by randomly choosing a fraction of the states in each batch to serve as starting points". What's the "batch" here?


**Time Spent Reviewing:**

5

---

> ### Author Response · Authors · 2021-08-10
> **Thank you for your review**
>
> Thank you for your thoughtful comments. We will take this feedback on board to improve the writing, and hope the clarifications below help with any confusion.
>
> For our baseline method in the deep RL experiments, the only differences to the published Muesli agent are documented in the hyperparameters in the appendix (a smaller batch size with a higher fraction of sequences from replay, which we found to perform a little more stably in some early experiments). We will clarify in the text that this is all we meant by ‘variant’.
> We agree that it is inconvenient to have details of the baseline relegated fully to the Muesli paper and the “Podracer architectures for scalable Reinforcement Learning” system paper. We will include more detail about the MPO loss in the main body, as well as more comprehensive implementation details in the supplementary material.
> We will also open source the code used for the tabular RL experiments.
>
> Self-consistency is certainly conceptually applicable to any RL domain, and in combination with most model-based baselines. We feel the empirical results in the paper give a fairly good coverage of some interesting benchmarks and use strong baselines, but are excited to see this type of method explored in more environments and with other baseline methods as well.
>
> *“Eq.7: How is G computed? Please give an explicit form rather than simply citing a paper if it's not elementary.”*
>
> We describe G_{k:K} in the text as “(K-k)-step bootstrapped value estimates constructed with rewards and states predicted by rolling out the model”. We can make this explicit as \sum_{j=k}^{K  - 1} \gamma^{j-k} \hat{r}_j + \gamma^{K-k} \hat{v}(\hat{s}_K).
>
> *“Sec 4.1: What's the model-free baseline? How's the value normalized?”*
>
> The model-free baseline is simply TD(0), as described in section 3.2. Every tabular method uses a TD(0) grounded value update using the real sampled transitions. The value error is computed as abs(v - \hat{v} / v) where v is the true value computed analytically; we will add this detail.
>
> *“(L.174) In contrast, the “direct” self-consistency update was able to accelerate learning, even compared to Dyna, for both MLE and value-equivalent models. I didn't see a significant advantage of SC-direct over Dyna with VE models.”*
>
> The advantage is indeed larger with MLE models than with VE models in our tabular experiments, though still present with the latter.
>
> *“Sec 4.1: Please provide a short description of the environments used (especially Sokoban).”*
>
> The Atari Suite consists of various discrete-action video games.
> Sokoban is a single player puzzle game where a player’s avatar can push N boxes (up down left or right) around a procedurally generated warehouse, attempting to get to a state where each box is situated on one of N target storage locations.
> Go is a two player abstract-strategy board game where the goal is to surround more territory on a NxN (in our case 9x9) board by placing black or white stones on the intersections of the board
> Please see appendix B.1 for further references about the environments; we will add some elaboration there.
>
> *“Fig.6 (a): 6 curves in the legend, 4 curves in plots? Fig.6 (d): The figure caption should match the titles of each subplot.”*
>
> Thank you for catching the error in fig 6a, which should have all 6 lines (for all the methods whose results are summarised in the barplots in fig 3a). We will regenerate the figure.
>
> *“The improvement of Muesli + SC over Muesli seems to be minor.”*
>
> As we note in the text, the overall improvements of the best variant we tested were quite modest. We find this itself to be an interesting result, and also note the value of the negative results for other self-consistency variants, and for a standard application of Dyna using the VE model.
>
> *Sec 4.2: Is the policy pi deterministic or stochastic?*
>
> pi is stochastic
>
> *“(L.257-259) We found that self-consistency updates were helpful, even though the model was neither used for value estimation nor policy improvement... I don't understand. In both VE aux and SC aux, the model is used to improve the value function. What does "the model was neither used for value estimation" mean?”*
>
> In the experiments shown in Fig 4a, the model is only learned as an auxiliary task; we use a standard actor-critic objective, and the critic value head is not updated using the model even when using SC. We will elaborate on this experiment setup in the supplementary material.
>
> *“L.273: "by randomly choosing a fraction of the states in each batch to serve as starting points". What's the "batch" here?”*
>
> The Muesli algorithm takes steps using the gradient of the loss calculated using a batch of transition sequences, some drawn from a replay buffer and some taken from the most recent experience gathered by the agent. For this experiment, we randomly sample some fraction of the states in this batch to use for the self-consistency loss.
>
> *“I didn't find where the authors discuss the limitations.”*
>
> While we do not include a specific section about limitations, we describe throughout various limitations of our method (its reliance on a grounded model & value update, the modest improvements observed in our experiments, the weakness to a poor reward model in S4.3).

---

### Official Review · Reviewer_nzxk · 2021-07-16

**Rating:** 7
**Confidence:** 3

**Summary:**

This work proposes to improve model-based RL methods by enforcing the model and value learned to be “self-consistent”, i.e. to satisfy the Bellman eqn where both the Bellman operator and value function are defined in terms of the model. The motivation behind this is that in classic MBRL, the model and value function are learned separately, and the value function is typically learned to be consistent with the model but not vice versa. The authors propose to enforce self-consistency by adding a "self-consistency loss” (L_SC) to the base loss and jointly learning the model and value functions. L_SC is the squared TD error with a stop-gradient on the target and is calculated using rollouts from the model. Although in MuZero/Muesli the model and value functions are learned jointly as well, the targets of the loss are all defined with respect to data from the true environment and there is not term enforcing the “self-consistency” between model and value functions.
Experimental evaluations on a selection of Atari games show that jointly updating the model and value functions with the additional L_SC term leads to improved sample efficiency as compared to Dyna and a model-free baseline. The authors also empirically study the search control problem.

**Limitations And Societal Impact:**

The authors do a good job of addressing the limitations of the method through empirical evaluations (e.g. the susceptibility of the value error to poor initialization of reward).

**Main Review:**

### Strengths:
The paper is very well-written and clear. Figure 1 is a nice schematic comparison of the closely related methods and helps put the proposed approach in context. The motivation makes sense based on previous work and the proposed approach is straightforward. The empirical evaluations are thorough in looking at the different variations and their effects.

### Weaknesses/Clarifications:
The related works section (particularly the first paragraph (Line 280-292)) could have more details and comparisons.

Question: Since the model is unrolled for many steps in the computation of L_SC and each step of the unrolling is not “grounded” to the true data, it seems that the errors in the model and the value function would interact and compound with more unrolling steps and gradient updates. Did you observe a challenge like this in your evaluations?

**Time Spent Reviewing:**

3

---

> ### Author Response · Authors · 2021-08-10
> **Thank you for your review**
>
> Thank you for the positive comments and feedback.
>
> We will try to make space for (or otherwise add to the supplementary material) a more thorough discussion of the comparisons to specific methods from the literature, to supplement the comparisons we make earlier in the text to the broader families of Dyna-like or VE-like algorithms.
>
> Unrolling the model for multiple steps does indeed lead to compounding errors as you suggest. In our deep RL experiments we found the self-consistency effect fairly similar for 2-5 model unrolls, after which performance degrades. We will add an ablation showing this effect to the supplementary material.

---

### Official Review · Reviewer_Vnna · 2021-07-22

**Rating:** 7
**Confidence:** 4

**Summary:**

This paper presents the idea of self-consistent reinforcement learning.  By enforcing "consistency" between value functions and approximate  (learned) models, a model-based learning algorithm can use synthetic experience generated from the approximate model.  This paper presents a few minor variations on this theme, and explores the results empirically.

**Ethical Concerns:**

There are no ethical concerns with this paper.

**Limitations And Societal Impact:**

I do not foresee any significant negative societal outcomes, beyond those inherent to all RL research.

**Main Review:**

In general, I liked this paper, although I wish the results were more compelling. I lean towards acceptance because I think this paper presents some good ideas, and that those ideas are worth exploring more.  The central idea that imagined experience can somehow be used to improve the model that it is generated from seems like a Really Good Idea, and I would like to see more work on it.

* Originality

As far as I can tell, this work is original.  While the RL community has explored many of these themes, this particular combination of ideas seems novel.

* Quality

The paper is of very high quality, blending novel technical insights, a solid positioning in the academic literature, and thoughtful experiments.  I appreciated the candid balance of positive and negative results - that's good scholarship.

* Clarity

The paper is generally well-written, although much of the more technical parts are difficult to understand how exactly all of the pieces fit together.

* Significance

The significance is mixed. While the central theme of the paper is potentially significant, and the technical development is solid, the experimental results are weak.  Often, the proposed methods perform only slightly better than competing methods.


**Time Spent Reviewing:**

1

---

> ### Author Response · Authors · 2021-08-10
> **Thank you for your review**
>
> Thank you for the broadly positive feedback and praise for the quality of scholarship!
>
> We fully agree that this is far from the final word in this space, and hope that the results and insights we present here (including the negative results for some methods, and the limited gains we report for the best approach we tried) will help direct future research.
>
> We will revise in particular the exposition of how VE models relate to self-consistency (see response to reviewer TCCY), and hope this will improve the clarity around how these pieces fit together.

---

> > ### Comment · Reviewer_Vnna · 2021-08-30
> > **Thank you**
> >
> > I have read the other reviews and your responses.  You have done an excellent job of responding, and other reviewers have articulated additional value in the paper that I did not see, so I am raising my score to a 7.

---

### Decision · Program_Chairs · 2021-09-27

**Decision:**

Accept (Poster)

**Comment:**

The paper proposes a self-consistency approach to model-based RL. The self-consistency refers to jointly optimizing the value function and the model such that (a variant) of the Bellman residual is minimized.

The reviewers are all positive about this work. They believe that it is an original work and written (mostly) clearly.  Most of their concerns are also addressed during the discussion phase, and there is no concern that prevents them from accepting this work. Therefore, I would recommend the *acceptance* of this paper.

There are some minor issues that can be improved though. Please refer to the reviews for more detail. I enlist a few of them:

- Improving the writing, especially in the Experiments section, in which some details are missing.
- Explaining and studying the effect of the number of unrolled steps in the computation of L_SC.
- Some of the reviewers believe that the paper isn't very reproducible. Please try to improve it in this regard, by providing more detail or releasing (some parts of the) source code.
- Better explanation of the relation between value equivalence and self-consistency.